# CoAware: Designing Solutions for Being Aware of a Co-Located Partner's Smartphone Usage Activities

Khalad Hasan*
University of British Columbia, Canada

Debajyoti Mondal†
University of Saskatchewan, Canada

Karanmeet Khatra‡
University of British Columbia, Canada

David Ahlström§
University of Klagenfurt, Austria

Carman Neustaedter¶
Simon Fraser University, Canada

## ABSTRACT

There is a growing concern that smartphone usage in front of family or friends can be bothersome and even deteriorate relationships. We report on a survey examining smartphone usage behavior and problems that arise from overuse when partners (married couples, common-law relationships) are co-located. Results show that people have various expectations from their partner, and often feel frustrated when their partner uses a smartphone in front of them. Study participants also reported a lack of smartphone activity awareness that could help decide when or how to communicate expectations to the partner. This motivated us to develop an app, CoAware, for sharing smartphone activity-related information between partners. In a lab study with couples, we found that CoAware has the potential to improve smartphone activity awareness among co-located partners. In light of the study results, we suggest design strategies for sharing smartphone activity information among co-located partners.

**Index Terms:** Human-centered computing—HCI design and evaluation methods—User Studies; User interface design

## 1 INTRODUCTION

Smartphones continue playing a pivotal role in our daily communications with family and friends [6, 42]. Smartphones not only enable seamless communication over long distances, but also allow access to information anywhere, anytime. However, there is growing evidence that people may overuse smartphones, both when alone and in the presence of others, i.e., in a co-located situation [3, 30]. Moreover, smartphones are designed as private and personal devices: the activities that take place on the screen can, when desired, easily remain completely unknown to co-located persons. Not being aware of a co-located person's on-screen activities can cause frustration and even anxiety [44, 46].

A significant amount of work has explored smartphone overuse and its consequences [3, 30, 49]. Much less attention has been devoted to designing solutions for co-located activity awareness which could mitigate the frustration associated with smartphone overuse and improve interpersonal communication. A few recent studies have attempted to increase smartphone activity awareness by helping people to be more aware of co-located people's smartphone activities, providing a rich shared experience, and even motivating people to initiate interaction with nearby persons [28, 44]. These studies suggested different strategies to raise awareness, such as using 'talk-aloud' to pass on what one is doing on the device [44] or to attach a second display to the back of the phone to show on-screen

*e-mail: khalad.hasan@ubc.ca
†e-mail: d.mondal@usask.ca
‡e-mail: karan26@student.ubc.ca
§e-mail: david.ahlstroem@aau.at
¶e-mail: first.last@email.com

activities to co-located individuals [28]. Though these solutions have the potential to increase smartphone activity awareness, they might not be appropriate in some common contexts (e.g., in social gatherings and in public places) and they might not be practicable due to the dependency on hardware instrumentation.

Social relationships can greatly shape the degree and nature of people's information sharing with other co-located individuals [10, 16]. For example, the information sharing patterns of people with their partner, parents, and children may be very different, and may even vary largely depend on the age of the individuals or the length of their relationship [5, 24]. To narrow down the focus, we concentrate on an in-depth investigation of various aspects of smartphone use by co-located partners who are married or in a common-law relationship, or in any other (romantic) relationship. We focus on such relationships because partners are often co-located for a substantial portion of each day and their mutual understanding is important for a healthy home environment [11, 21, 43, 50]. Couple relationships are already very nuanced and complex and include many aspects, e.g., closeness, connectedness, interpersonal trust, and perceptions of empathy. This makes them even challenging to study on their own as a focal relationship [10, 21, 41].

We first conducted a crowdsourced study aiming to explore the following research objectives: 1) understand people's smartphone habits when being co-located with their partner, 2) gain insight into the concerns that people have about their partner's smartphone usage, 3) understand the rules and privacy issues partners have regarding their co-located smartphone usage, and 4) explore the strategies that people take to become aware of their partner's smartphone on-screen activities. Our results show that people often use smartphones while co-located with the partner, which sometimes leads to anger and frustration. We also found that people often need to respond to their partner's queries about their smartphone activities. Most people share information truthfully, although the details of the shared information vary widely between apps. Furthermore, many people feel that they are not fully aware of their partner's smartphone activities when co-located. This lack of awareness can lead to unpleasant situations. Some strategies such as 'talk-aloud' are being used to be aware of others' smartphone activities, yet there remains a lack of expressive tools to support smartphone activity awareness.

Guided by these findings, we explored ways to increase smartphone activity awareness among co-located partners. Our goal was to investigate smartphone-based solutions to help partners become aware of each other's smartphone activities and to help them improve their interpersonal communication. We developed a smartphone app, CoAware, that enables users to create co-located smartphone usage awareness by sharing the names, categories, or screens of apps being used by one's partner. Additionally, CoAware enables partners with ways to send notifications to each other such that they might motivate the partner to reduce co-located smartphone usage. We continued with an in-lab study with couples who explored and provided feedback on the features offered by CoAware. The results revealed that high-level information, such as sharing app names and sending notifications, is useful to provide co-located smartphone usage

awareness; however, low-level information about phone usage (e.g., screen sharing) and allowing co-located partners to control a different person's phone were seen to be less necessary and sometimes overbearing. We also found that awareness of a partner's smartphone activities was not desired by all participants. Some participants were satisfied with the level of information they knew about, and some were fine relying on social protocols with their partner to remedy challenging situations. Thus, design solutions should not be thought of as a one size fits all approach.

## 2 RELATED WORK

### 2.1 Smartphone Overuse and Reduction

Smartphone overuse and smartphone addiction are active areas of research that examine people's smartphone usage behavior. Prior work showed that the overuse of smartphones can lead to a decrease in productivity in workplaces due to phone use during work hours [12], hamper family relationships, and even cause domestic violence [31]. Researchers have investigated ways of detecting smartphone addiction based on users' smartphone usage behavior. For example, a recent study [42] identifies lifestyle and social media related apps to be associated with smartphone addiction. There are commercial apps that track users' smartphone and computer usage and offer summarized information allowing users to be more productive in their daily activities [39, 47]. Researchers have also explored solutions to provide people with their desktop computer usage information [23, 32], and non-work related web access information [37]. Despite such systems, further exploration is needed to examine how to apply this knowledge to design persuasive smartphone interfaces to promote co-located engagement for improved interpersonal relationships.

### 2.2 Information Sharing through Digital Devices

Sharing digital devices and accounts are common practices amongst household members, yet this topic has received less attention from an awareness perspective. Instead, the topic has mostly been motivated by the pros and cons associated with device sharing. Sharing devices can be viewed as an all-or-nothing approach for sharing information [29, 40], which gives rise to privacy and security issues [5, 27]. Studies also showed that device sharing concerns depend on the user's relation to the other user as well as the types of data being shared, which suggests the need for better privacy and security models for device sharing [18, 20, 29]. The intricate ways that couples communicate [10] and their need to have a nearly continuous connection have gained deep attention in the literature [21, 43, 50]. This motivated context sharing among people in close relations. Prior research [8, 15, 16, 33] showed how contextual information sharing (heart rate, distance from home, etc.) can be leveraged to make partners aware of each other's context and activities. Both Buschek et al. [8] and Griggio et al. [16] observed that context-awareness improves the sense of connectedness and pointed out interpretability and privacy concerns that may arise due to inferred additional information from the given context. While context sharing helps people in close relationships to be aware of each other's activities, smartphone addiction and overuse of other digital tools may create an awareness barrier even when people are present in the same context.

### 2.3 Smartphone Activities among Co-located People

A number of studies analyzed smartphone distractions during group activities [28, 34, 36]. Ko et al. [34] developed a smartphone app that allows a group of co-located users to simultaneously lock their smartphones during activities such as studying and chatting. Jarusriboonchai et al. [28] proposed an approach to communicate user's activities on the backside of a smartphone, by displaying the icon and name of the app being used, which raised privacy concerns. Oduor et al. [44] examined people's use of smartphones in the presence of family members in the home. They found that people often feel that the smartphone usage of their co-located family members is

non-urgent. They feel ignored when not knowing their true activities. However, many fundamental questions are yet to be explored: When, where and how often do such problems arise? Are couples interested in knowing each other's smartphone activities? If so, then to what extent? How often do couples share activity information? With how much detail and how truthfully? Does the relationship, privacy, trust, or smartphone app play any role? We cover these aspects and more.

### 2.4 Usage-Aware Co-Located Smartphones

Though there has been substantial work on different sensing solutions that allow users to track the state of the device [22, 48], very little is known about how to detect two co-located smartphones and how to share information between them. Prior research showed that on-device sensors (e.g., tilt) could be used to make a smartphone context-aware of its state (e.g., orientation) [22], and usage context (e.g., resting on a table) [48]. Beyond sensing a device state or context, researchers explored external sensing solutions to enable smartphones to track surrounding activities and environments [9, 17, 19, 35] and their social acceptance [1], but in limited contexts. To date due to the lack of advances in sensing solutions, such an approach has received little attention in the context of using the space for co-located collaborative interactions for promoting interpersonal engagement. CoAware provides a way of pairing co-located smartphones within 200 meters without requiring any WiFi hotspot or smartphone data connection and thus allows co-located users to connect with each other's devices.

## 3 STUDY 1: EXPLORATION OF CO-LOCATED SMARTPHONE USAGE AND ACTIVITY AWARENESS

We started our exploration by conducting a crowdsourced study investigating people's smartphone usage, rules, trust and privacy concerns, and activity awareness when they use smartphones in the presence of their partner. Prior research has shown that crowdsourcing platforms, such as Amazon Mechanical Turk (AMT), are popular and convenient tools for conducting user studies and collecting reliable data [2]. We used AMT to run our study.

### 3.1 Online Survey

We created an online survey with 58 questions to collect data from smartphone users. The survey contained five sections: i) 18 questions to collect demographic information about participants and their partners (e.g., age, nationality, gender, education, and household conditions); ii) 15 questions about smartphone usage (e.g., how often, where and what types of apps), usage rules in the household, and privacy-related issues; iii) 9 questions about trust-related issues that arise when people share their smartphone usage activities with their partner and other family members; iv) 5 questions targeted at smartphone usage behavior and habits when co-located with the partner; and v) 11 questions regarding the awareness of the partner's smartphone usage and possible strategies used to share usage related information with a partner. In total, we used 15 open-ended questions, 26 single/multiple-choice questions, and 17 5-point Likert scale questions. Most of the open-ended questions were used to collect descriptive responses about co-located smartphone usage where the Likert scale questions were designed to quantify results and to obtain shades of perceptions regarding issues on smartphone usage.The single/multiple-choice questions were primarily used to collect demographic data.

### 3.2 Participants and Study Procedure

We posted our survey as a Human Intelligence Task (HIT) to AMT (with a $1.00 compensation). We specified two qualifications for participants: a minimum of 70% approval rate and a minimum of 50 previously completed HITs. We also set the following requirements for the workers: they (a) must own a smartphone, (b) be either married, or in a common-law, or in a partner relationship, where

(c) the partner must also own a smartphone and (d) currently live in the same household. In total, we collected 109 responses in seven days. We subsequently removed 31 responses which contained one or more unanswered questions and/or invalid answers. Consequently, we analyzed data from 78 participants (34 female, 44 male). On average, participants took 25 minutes to complete all the questions.

### 3.3 Data Analysis and Results

We applied a thematic analysis on the qualitative data where two researchers separately went through all the comments to perform open coding. Later they consolidated and reconciled codes into a common code set. Self-reported quantitative data were analyzed using standard statistical methods such as mean and standard deviation.

#### 3.3.1 Demographics and smartphone usage

The majority of our participants were from two age ranges: between 24 and 34 years (28 participants) and between 35 and 44 years (29 participants). Three participants were aged between 18 and 24 years, nine between 45 and 54 years, and nine between 55 and 64 years. Only two participants were 65 years or older. Fifty-five participants were from the USA, 23 from India. On average, our participants had been in their relationship for 13.3 years. Participants and their partners had been using smartphones for 8.5 and 7.9 years, respectively. Participants reported using smartphones an average of 3.3 (SD=2.0) hours per day.

Participants indicated using some categories of apps more often than other categories: 90% used communication apps (e.g., email, text message, skype, phone calls) at least once a day and 83% used social media apps (e.g., Facebook, Instagram, Snapchat) and 85% used the Internet (e.g., reading news, hobby-related browsing, banking) at least once a day. Only 19% of the participants used location-sharing apps (e.g., Glympse, Life360, Find My Friends) once a day and only 29% used health-related apps (e.g., fitness tracking, sports, or medicines apps) at least once a day.

#### 3.3.2 Co-located smartphone usage

Participants reported that they are co-located with their partner for a considerable amount of time (mean 5.9, SD=3.6 hours/day excluding sleeping time) and that they often engage in collaborative activities with their partner, such as cooking or watching movies (mean 2.2, SD=1.4 hours/day). In response to an open-ended question, participants reported various reasons for using their smartphones when co-located with the partner. In total, we analyzed 167 coded responses which can be categorized into the following five broad categories: 1) communication/ socialization with friends or family members (38% of the responses), 2) work-related activities (20% of the responses), 3) checking information and updates for own interest (20% of the responses), 4) finding information for a purpose shared with the co-located partner (13% of the responses), and 5) personal entertainment (9% of the responses). These results are similar to earlier qualitative results [44] which showed that people use smartphones in the presence of their family members to check notifications, find information, and fill time when they are bored. Our results also revealed that co-located smartphone use frequently happens at home (44% of the responses), mostly in the living room and bedroom. Many participants talked about using their phones at home when co-located with their partners. "... *watching a movie on tv at home and he was upset that I checked my phone.*" [P20, female, relationship 27 years]

Other common places for co-located smartphone use include restaurants (18% of the responses), public spaces such as in shopping malls or parks (24%), during social gatherings (7%), and inside cars (7%). All participants reported frequent occurrences (at least once a month) of their partner expressing concerns regarding their co-located smartphone usage. "*I pulled out my phone just to go on it before the food came and he complained.*" [P1, female, relationship

4 years] || "*We were in bed together and not really paying attention to what she was saying.*" [P16, male, relationship 21 years]

We coded a total of 114 responses regarding the situations (places and activities) when this had happened: at mealtime, either at home or in a restaurant (29% of the responses), while watching tv/movies together at home (16%), in public places, such as in a shopping mall (15%), in the bedroom at bedtime (13%), during on-going conversations (10%), and in some other situations such as at a social gathering or while in the car. Two participants from India responded that it had happened while being in a temple.

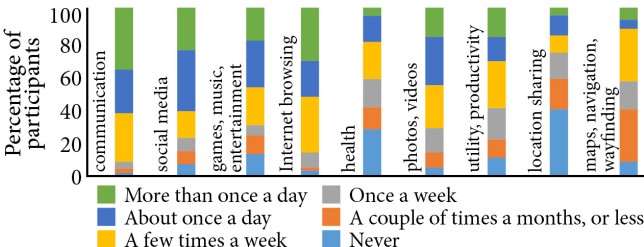

Figure 1: Participants' self-reported smartphone application usage frequency when co-located with their partner.

Participants' concerns were mostly related to the lack of attention to what they had expected their partner to concentrate on during a conversation or other activity. Sometimes they were concerned about disregarding family time and social engagement (especially when surrounded by family or friends in social gatherings). Several participants mentioned that their failure at paying attention sometimes led to frustrations and tensions between them. "*When we sit together and talk to each other in our living area at home, I go through messages in WhatsApp. That time my partner gets irritated, thinking that I am not listening to him.*" [P10, female, relationship 21 years] || "*During a dinner at his friend home... I was using my phone continuously to text my friends. He signaled me not to use the phone at a get together because it seems odd when I am not involving in the event. I keep on texting my friends he raised and fought with me.*" [P21, female, relationship 10 years]

Participants reported that their partner expressed concerns about their co-located smartphone usage primarily due to disruption in their quality time, and sometimes expressed anger and annoyance. On average, participants reported that they spend 2.3 hours a day on their smartphones while co-located with their partners. For each app category, at least 30% of the participants who use those apps more than once a day reported to use them less frequently while co-located with their partner.

We also asked participants about the apps that they often use when they are co-located with the partner. Figure 1 shows the results. We observe that they frequently surf the Internet (e.g., reading news, browsing, banking), use communication apps (e.g., email, text message, Skype, phone calls) and social media apps (e.g., Facebook, Instagram, Snapchat). However, they rarely use health-related apps (e.g., fitness tracking or medicines apps) and location sharing apps (e.g., Glympse, Life360). The results suggest that people prefer to use communication and other related apps to connect to families and friends when co-located with their partner.

#### 3.3.3 Rules or mutual understanding on smartphone use

We asked participants questions about rules or agreements set in the household to reduce co-located smartphone usage. More than one-third of the participants (35%) mentioned having some house rules. The rest (65%) said they did not have any formal agreement, yet they shared a mutual understanding with their partner. "*We're responsible and adult enough to know when it's time to use the phone or not.*" [P52, male, relationship 11 years]

The participants who reported to have rules or agreements (43 coded responses) for smartphone use had rules based on either locations or situations. Mealtime (33%), family time with kids (21%), collaborative activities (16%), bedtime (12%), social gathering (7%) and driving (5%) were some contexts where the rules restricted smartphone use. *"We agreed to not use our smartphones during dinner unless it's an emergency."* [P46, male, relationship 26 years]

Often the rules or agreements were set to ensure quality time within the family and in social gathering: *"I am in agreement with him that we do not use our phones when it's quality time for us to be together or when we're with others in a social situation, unless everyone is using them, too, for some reason (like looking up some info or playing a game together)."* [P58, male, relationship 8 years]

The rules also came from self-realization of being disconnected: *"Once me and my spouse was continuously using the phone when we were at home ... we realized that we didn't speak to each other. That moment we decided not to use phones unless an emergency when we both are together."* [P21, female, relationship 10 years]

We asked the 51 participants, who did not have any rules, how they would feel about creating them. Fifty percent of these participants welcomed the idea of having some rules for ensuring proper engagement with the partner. Twenty-five percent expressed being somewhat neutral about agreeing on rules. The remaining 25% opposed the idea of agreeing on rules. They did so as they felt that rules would intrude on their smartphone activities or that it may be an "overkill" between adults who should be able to act on their own accord. Some stated that they shared mutual respect not to use smartphones in certain situations and do not need any rules. *"We should come up with guidelines for smartphone usage that would make me feel better about our communication with one another."* [P37, female, relationship 14 years]

The average length of relationships for the participants who have no rules was larger (avg. 14 years) compared to the participants who reported to have rules (avg. 10 years). Binomial Logistic Regression showed no significant difference in gender and relationship length between these two groups.

We asked participants whether they have any rules regarding smartphone use for other family members, excluding themselves, such as their parents (e.g., an older adult living at home with their adult children), teenage children, or younger children. Most participants mentioned that they do not have any rules for other adults in the home as they are responsible adults who do not use smartphones frequently. *"There is no rule as my mom is an aged woman and did not use phones every day."* [P39, male, relationship 12 years]

However, 40 participants with children have strategies and rules to control the child's smartphone usage. For instance, out of 52 responses, 37% of the responses were about time-based restrictions (e.g., no more than 30 min per day), 23% were about content-based restrictions (e.g., only for games and watching YouTube videos), 21% were location-based restrictions (e.g., not at the dining table, in the bedroom or washroom), and 6% were about age-based restrictions (e.g., no phone before 8 years). Such "no phone" policies were primarily set to ensure that the children were engaged in more purposeful activities and to ensure they spent enough time with family members. Typical responses were: *"We do not allow our sons to use their smartphones in private such as their bedroom or bathroom. We also have their settings configured so their phones may not be used between 10pm and 6am."* [P46, father of 2 children]

### 3.3.4 Strategies to reduce co-located smartphone usage

We also asked our participants whether they know of or used any apps or other tools to reduce smartphone use in co-located situations. The majority (67 out of 78) mentioned that they do not know of any such solutions that could either help them be more aware of each other's smartphone activities or help to reduce co-located smartphone usage. The remaining participants (11) mentioned that they are aware of apps to restrict usage time. They mentioned using iPhone's Screen Time [26], Night Mode [25], Offtime [45] to track their daily smartphone usage activities and to limit smartphone app access after a certain amount of time.

We used a 5-point Likert scale to collect participants' opinions on the importance of using apps or other strategies to reduce co-located smartphone usage. We observed that younger participants felt that it was more important to have apps or strategies in comparison to the older participants. Out of 28 participants in the age range 25 to 34 years, 17 participants expressed high importance (rating 3 or more) of having such apps or strategies, whereas only 25 of 50 participants in the age range 35 to 65+ years expressed high importance.

### 3.3.5 Sharing information, privacy, and trust

About 74% of participants said that they told their partner what they were doing on their smartphone when co-located at least a few times a week. We also asked them how truthful they are when sharing information. Twenty five out of 34 females and 19 out of 44 males said that they share accurate/truthful information about their smartphone activities with their partner. Participants who said they told the truth commented that they do not have anything to hide from their partner and do not want to lose the trust. *"We value honesty in our relationship, not that we do anything shady on our phones, but if we did, I would immediately inform her of anything I did, and vice versa."* [P52, male, relationship 11 years]

On the other hand, participants who said they did not always share accurate information with their partners did so because they were trying to safeguard their privacy or ensure personal boundaries. For instance, some participants mentioned that they are not comfortable sharing financial information, business matters, photos, videos and things that they search on their phone. This is due to the sensitivity of this information, and sometimes to maintain personal space [50]. *"I might be slightly embarrassed about the random things I look up."* [P33, female, relationship 20 years]

In a follow-up 5-point Likert question (5=very confident to 1=not confident at all), participants were asked to indicate their confidence level about whether their partners tell true information about their smartphone activities. Both male and female participants had strong confidence that their partners share accurate information with them, which reflects their average score of 4.45 (SD=0.84) and 4.68 (SD=0.79), respectively. Only six participants gave a rating of 3 or below and expressed their past experience of finding their partners not being truthful. This experience could consequently create an impact on their level of trust in the future. *"About 9 months ago my partner expressed that due to past cheating by previous partners, she felt paranoid when I was using my phone to chat with other people."* [P8, male, relationship 1 year]

### 3.3.6 Smartphone usage awareness

We examined participants' awareness about exactly what their partner is doing on their smartphone and how interested they are in knowing what their partner does on their smartphone. Seventy eight percent of the participants responded that they are not fully aware of their partner's smartphone activities. In some cases, participants reported that this lack of awareness triggered misunderstanding among the co-located partners as their partners make assumptions based on their smartphone activities. A potential reason for such an assumption could be the limited information that can be seen from a distance about a person's usage activities. Similar results were found by Oduor et al. [44] who reported a lack of smartphone activity awareness among co-located family members.

We included questions on the common strategies for sharing activity awareness with co-located partners. Participants reported that such awareness was often achieved by asking questions of their partners where they responded verbally or showed their screen to their partner. This action sometimes led to frustration and anxiety

among partners. "*I normally just ask what he's doing (especially if he laughs!) and he'll always tell me.*" [P1, female, relationship 4 years] || "*My partner usually gets aggravated when I ask what he is doing, because usually, he is trying to figure something out on his phone.*" [P11, female, relationship 15 years]

In exploring how interested participants were in knowing the partner's smartphone activities, we found that male participants were more interested (66% of the males were interested) in knowing what their partner is doing on the smartphone than female participants (58%). On the contrary, in a question asking about their partner's interest in knowing what they are doing on their smartphone, we found that 86% of male participants reported their partners to be interested in knowing their activities, whereas 68% of the females reported the same. We observed a trend that this interest decreased gradually with the increase of age range. In the age range 25 to 34 years, 82% expressed interest, whereas in the age range 35 to 44 years only 70% showed interest.

### 3.3.7 Co-Located Content Sharing

We collected information on level (details vs. abstract) of smartphone activity information that the participants are comfortable to share with their partner and the level of information that they would like to receive from their partner. We provided them with three different levels that they could choose from for sharing or receiving: (i) detailed information (e.g., chatting with "Alex" in Facebook), (ii) an app's name (e.g., using Facebook), and (iii) activity information (e.g., playing games). Additionally, they could write any other abstractions that they might be comfortable with. We collected 119 coded responses for sharing and 120 coded responses for receiving level of information as they were allowed to select multiple levels.

Many participants are comfortable with providing very detailed information to their partners (37% responses), whereas others reported preferring to share only the app name (34% responses) or general activity information (29% responses). The other participants reported only feeling comfortable with providing less or no information at all. We also found that the preferred sharing level varies across apps. 37 participants mentioned that they share details when using communication apps whereas only 19 participants share details while they are browsing the Internet. We observed similar results for receiving information from their partner. Many participants indicated that they would like to receive detailed information from their partners (34% responses), whereas others expressed to receive only the app name (36% responses) or activity information (29% responses). The other participants (only 1%) reported feeling comfortable with receiving any level or no information at all.

We further asked participants to provide examples where they share smartphone usage information with different people (e.g., partner, family members). We observed that it is common to share different levels of information with different people: "*I would give less details based on how well I know the person. My partner and family get more information that colleagues.*" [P57, male, relationship 2 years] || "*To my partner, I share all the information; to my family members, I share only app name or activity name*" [P53, male, relationship 7 years]

### 3.4 Discussion

Results from the survey revealed that people use smartphones in the presence of their partner even though their partner expressed concerns about the usage. In general, people can see when partners use smartphones in front of them, but exactly what a partner is doing on the phone cannot be easily inferred from an observer's viewpoint (also found by [28]). Our work builds on prior work by illustrating the locations and activities in which this occurs, the rules people have setup to help mitigate issues, and how they feel about sharing usage information. Participants reported that co-located usage and asking about their partner's activities sometimes triggers

aggravating situations. However, there are no known technological solutions available that would help them share their smartphone usage information with their partner while offering some degree of privacy to increase activity awareness and create shared experiences in co-located contexts.

These findings motivated us to think of a means to improve people's awareness about their partner's phone activities while co-located by supporting different levels of information (details vs. abstract), such that they can make informed decisions about how to handle the situation. Additionally, prior research showed that improved awareness could help couples to stay in sync [4] and be motivated to limit phone usage and, thus, improve the quality of domestic life [38]. We recognize that awareness of a partner's smartphone activities was not desired by all participants. Some participants were satisfied with the level of information they knew about, and some were fine relying on social protocols with their partner to remedy challenging situations. Thus, a natural progression warrants an examination of design solutions that might work for people who were more interested in additional knowledge of what their partners were doing on their phone in a hope to improve social interactions. Prior work showed that smartphone activity awareness can be achieved with additional hardware instrumentation such as attaching an additional display on the backside of a phone to provide cues to smartphone activities [28]. Results revealed that users did not feel comfortable using it and were unwilling to reveal app related information due to privacy concerns. Instead, we developed an app-based solution on an unmodified smartphone to provide new opportunities to increase activity awareness among partners. With our design, CoAware, users can intuitively share activity information with various granularity levels to help avoid privacy concerns.

## 4 THE DESIGN OF COAWARE

Informed by our findings, we designed a smartphone app, Co-located Awareness (CoAware), intended for sharing smartphone usage information between co-located partners.

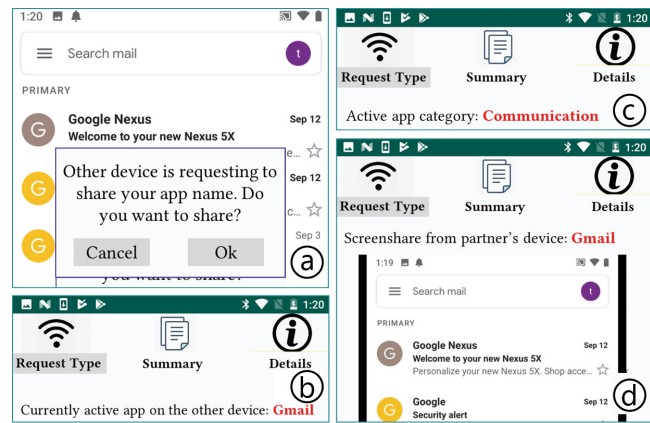

Figure 2: (a) A connection initiates with a request to gain access to the partner's device; Once the access is gained, the partner can see (b) the app name, (c) the app category, and (d) a screen showing the screen content of the app that the partner is viewing.

### 4.1 CoAware Features

CoAware was designed to share users' smartphone usage information such as the number of times an app has been launched, the duration it has been used for, and the time it was initially opened. We used a foreground app checker external library [51] that allows access to smartphone app usage information. In addition, we developed a solution to directly share screens from one smartphone to

another via WiFi direct. Based on these capabilities, we developed three techniques to share app usage information between two co-located smartphones with CoAware. As our survey results showed that people prefer to share information by varying degree of details about app usage, we designed the app to have three different levels of access; from very limited information which users may be more comfortable sharing (e.g., an app category) to very specific information (e.g., viewing the screen) that could possibly be more privacy intrusive. Thus, users can choose what level of sharing they and their partner are comfortable with. The specific levels are:

**App Category**: This access level provides users with a high-level view of app usage information, where only the types of apps being used are shown and not the app names (Figure 2c). For instance, apps that are used for contacting other people (e.g., email, text message, skype, phone calls) are mapped to and labeled as "Communication." Commonly used apps are categorized into different labels.

**App Name**: In this access level, CoAware tracks the name of a running app on the co-located phone (Figure 2b). The app name is displayed on the other phone.

**Screen Share**: In this access level, CoAware captures images of a phone's screen and transfers it to the co-located phone every 50ms. In this way, co-located users are aware of the exact on-screen activities of each other (Figure 2d).

With CoAware, when two devices are co-located, one device sends a connection request and asks permission to access App Name, App Category or Screen Share information. The receiving device shows the request in a pop-up (Figure 2a) where the user of the device can accept or reject the request. If accepted, the sender device gains access to the app name, app category or the device screens of the other device. It also starts logging the app usage information (e.g., running app) on the receiving device. CoAware provides three notification strategies to allow co-located partners to send information through the app. We wanted to provide various levels of information exchange and control.

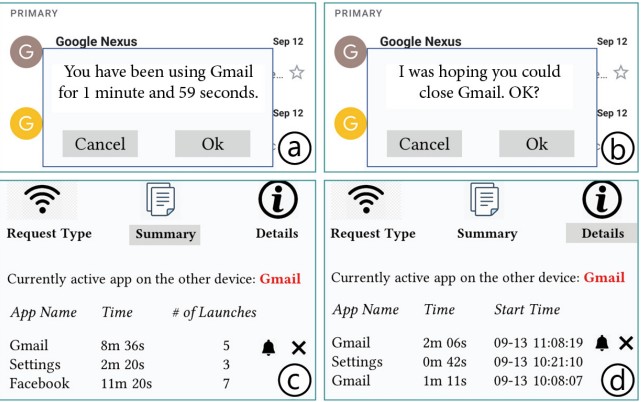

Figure 3: CoAware (a) sending usage statistics, (b) a close request, showing (c) a summary, and (d) detailed information about apps used since the connection was established.

**Message**: Users can send a preset or custom message to their partner. Examples of preset messages are "It feels like you've been using Gmail for a while now, can we talk instead?" and "Hey, it's me. How are things going?" Custom messages allow users to type anything in a text box and send the text to their partner. We included this possibility to offer flexibility in terms of how users like to communicate with their partners. This messaging feature is similar to sending a text message; however, we hoped that preset suggestions for messages might help to create courteous exchanges between partners and not heighten tensions. This reflects findings from our survey where some participants said they would gently

ask their partner about their smartphone usage if they felt it was inappropriate.

**Usage Statistics**: Users can send app usage statistics such as "You have been using Gmail for 4 min and 54 seconds." to their partner (Figure 3a). Such messages are created using the duration of the longest-running app among the currently running ones, since the longest-running app is often likely to keep the user engaged for a longer period of time. This reflects findings that some participants did not realize how long they were on their phone in the presence of others; thus, some additional awareness information could be useful to regulate behavior on one's own.

**Close Request**: Users can send a request to close the currently active app that their partner is interacting with. The partner sees a prompt such as "May I request you to close Facebook?" or "I was hoping you could close Gmail. OK?" (Figure 3b). The partner can cancel the request and continue using the app. If the partner agrees (tapping the Ok button), the app shows a 30-second countdown timer. Rather than closing the currently running app instantly, the time is given to let the person finish the current activity [7]. When the 30 seconds are over, CoAware closes the currently running app. Our goal here was to make the app closing somewhat graceful and delayed and less of an immediate interruption. This reflects findings from our survey where, again, some participants said that they might ask their partner to change their behavior on their smartphone. We recognize that causing actions to occur on someone else's phone may come across as being strong or overly assertive to some people. We wanted to explore this idea to see how people would react to it in further studies.

Using the Summary tab, one can see the usage statistics for the apps. The summary includes the app/category name, the total time the app or category has been used and the number of launches since the current connection was established (Figure 3c). If the access level App Name or Screen Share is given by the partner, then one can see the app names. The access level App Category only allows one to see the app categories. Using the Details tab, one can see more detailed information about individual apps or app category launches (depending on the access level), such as the name, launch time, and duration of use (Figure 3d). Overall, we recognize that not everyone will find the features we propose in CoAware to be useful. Some may find them to be overbearing, some may find them to be not needed at all, and some may find them to suit their needs well. This is as expected and purposeful such that we could explore our design ideas more and see what reactions participants would give with a fully working system that provides such options.

## 4.2 Implementation Details

CoAware was built on Android SDK 4.4 and leverages smartphones' WiFi Direct to establish peer-to-peer connections between two smartphones. We used WiFi direct as this technology allows two devices to connect directly without requiring them to connect via Wi-Fi routers or wireless access points, thus enabling sharing information between co-located users in any location (e.g., at home, park). Prior research has shown that smartphone activity can be shared with others by instrumenting the device (e.g., attaching an additional display to the back of the device) [28] which can raise privacy concerns due to the visibility of private content in some common contexts (e.g., public places). Hence, we designed an application solution that does not require any additional hardware instrumentation.

## 5 STUDY 2: EXPLORATIONS OF COAWARE

We conducted a study in a lab setting as an initial attempt to get feedback from participants about CoAware. Note that the lab study was designed to gain insights about the benefits, privacy risks and opportunities of CoAware and generate directions to guide future researchers while designing apps to increase activity awareness among co-located people. We investigated users' feedback on the three

access levels for sharing app information amongst co-located partners and explored their opinions on the three notification strategies provided by CoAware. Additionally, we collected participant feedback on privacy issues and on how CoAware creates awareness and we asked for general feedback on CoAware's features. Naturally, we could have explored our ideas using a field study where participants from various socio-cultural backgrounds could have tried out CoAware over a prolonged period of time. We did not use this approach given that CoAware is still at an early design stage. Field studies bring the risk of participants not trying out all of the features within a design. We felt it was more reasonable to gather initial participant feedback such that the general ideas presented in CoAware could be assessed to understand which may hold the most merit. Then, either CoAware or other applications like it could be created and explored through longer-term usage. The caveat is that our study does not provide generalized results across a range of real-world situations. We do acknowledge that a further field study is required to evaluate the app in the wild.

## 5.1 Participants and Procedure:

We recruited 22 participants (11 couples) from the local community (a large city within North America) to participate in the study. Two participants were 18-24 years old, 11 were 25-34 years old, 4 were 35-44 years old, 2 participants were 45-54 years old, and 3 participants were 55-64 years old. All participants were smartphone users and have been in their partner relationship for an average of 9.5 (SD=5.7) years. None of them had experience using tools to reduce smartphone usage or to support the awareness of someone else's smartphone use.

We used two smartphones, Google Pixel 3 and Google Nexus 5, for the study. We first showed participants how to use CoAware. Next, participants were given the following two tasks to complete:

**Establish a connection**: One person (sender) sends a connection request and the other person (receiver) accepts it.

**Access information**: The sender accesses the app name, app category, and screen on the receiver's device while the receiver 1) browses information on an e-commerce website (Amazon) to find a suitable camera costing less than $500, 2) finds a rumor/gossip about their favourite actor/actress, and 3) plays a game of their choosing. Once the tasks are completed, the participants switch roles as sender and receiver and repeat the tasks.

We then used a questionnaire to collect their opinion on the access levels and notification strategies to create co-located smartphone usage activity awareness, privacy concerns related to CoAware, and design suggestions to improve the app. We asked participants close-ended questions using 5-point Likert scales regarding (Q1) the usefulness of the three access levels in creating awareness about their partner's smartphone activities, (Q2) the usefulness of the three notification strategies to motivate them in reducing co-located smartphone usage, (Q3) their comfort level when receiving a notification from their co-located partner (for each of the three notifications strategies). Additionally, they were asked open-ended questions about privacy and awareness issues. In the end, we also asked to provide feedback and suggestions regarding CoAware's features. A session lasted approx. 40 min. in total. We used thematic analysis to analyze qualitative responses where we iteratively reviewed the responses to look for main themes.

## 5.2 Results

(Q1) Figure 4a shows the mean rating on how useful the three access levels were to create awareness of partners' activities. We found a mean rating of 3.91 (SD=0.53) for App Name, 3.0 (SD=0.93) for App Category, and 4.14 (SD=0.89) for Screen Share. We observed that App Name and App Category were rated more useful in creating an activity awareness than App Category. (Q2) Figure 4b shows the

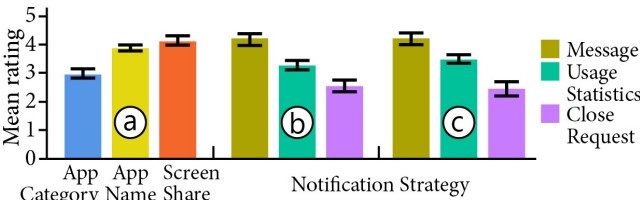

Figure 4: Mean rating for (a) the usefulness of the access levels, (b) creating activity awareness with notification strategies, and (c) the usefulness of the notification strategies.

mean rating for the usefulness of three notification strategies to motivate reducing co-located smartphone usage was 4.18 (SD=1.05) for Message, 3.27 (SD=0.83) for Usage Statistics, and 2.55 (SD=1.18) for Close Request. (Q3) Figure 4c shows that participants were more comfortable receiving a notification with Message (4.18, SD=1.05) and Usage Statistics (3.5, SD=0.9) than with Close Request (2.5, SD=1.3).

In the open-ended questions about privacy and awareness issues, participants expressed that CoAware would be helpful to maintain their time commitment to each other, create more smartphone usage awareness so that they do not interrupt their partner during an important ongoing activity, and that Screen Share would help them during co-located collaborative activities such as sharing information with others. "*[CoAware] can be very useful for creating awareness as it allows us to check what other has been doing, especially when he is on phone for a long time.*" [P7, female, relationship 5 years] || "*The app will help when we want to show something to each other but sitting different places in a room.*" [P3, male, relationship 5 years]

Additionally, participants mentioned that there are other potential use cases for CoAware (e.g., sharing information with their partner, monitoring their children's smartphone usage). "*Sharing feature is useful as I can show photos and videos to my wife; I can share the game that I am playing to my son.*" [P18, male, relationship 13 years]

Participants also expressed privacy concerns regarding the Screen Share. For instance, two females mentioned that they used some apps to track their health-related issues which they might not feel comfortable sharing. Others wanted to have a personal digital space away from their partner which they did not want to be intruded in, as this might create stress and tension in family life. "*It will hamper my privacy, I may not feel comfortable at all for sharing screens of my messages and emails*" [P4, female, relationship 5 years]

Participants provided suggestions to improve CoAware. For example, instead of showing pop-ups, they suggested using standard notifications that commonly appear at the top of the screen. Six participants also suggested that partners should be allowed to only send a fixed number of notifications within a certain time (e.g., 10 notifications per day). Some participants wanted more notification styles and strategies or more statistics to better motivate the partner to engage with them. Two participants also felt that instead of sharing the entire screen with their partner, a blurred image or custom screen area could be shared as this would protect privacy.

## 6 DISCUSSION AND DESIGN RECOMMENDATIONS

Through our studies, we were able to uncover a range of ways that applications like CoAware should be designed, based on its strengths and weaknesses.

**Ensuring personal digital space**: The participants' lower comfort ratings when using Screen Share illustrate that there is often a personal digital space among partners which they typically want to preserve. Thus, we suggest that applications like CoAware should focus on sharing higher-level or more abstract information (e.g., the app name) instead of sharing very detailed information such as the

screen content. Low-level information akin to what we provided as one feature within CoAware is likely too much for many people and could inadvertently create greater tensions between partners.

**Level of access**: Participants expressed concerns about using the Close Request feature within CoAware as it takes some control over a partner's phone and may disrupt their on-going activities. This, again, could create further tension between partners. Instead, participants felt that solutions that alert others of what they may want to change in their own behavior, rather than take control, would be more acceptable. Thus, when designing apps for communication between devices, it is important to carefully consider how much control one should have over another person's device.

**Notification Strategies**: Participants found pop-up notifications to be distracting and somewhat overbearing. Thus, we suggest that applications like CoAware use standard notification mechanisms that are already found on smartphones. Pop-ups notifications can be distracting and the forced change in activity (e.g., switching from a game to notification UI) could create new frustrations.

**Determining whether the phone is in use**: Sometimes a phone may remain active although the user may not be engaged with it. This means that usage information provided to one's partner may not be accurate. One possibility could be to rely on information about the user's on-screen taps in combination with information about the running apps to determine usage history.

Study 1 provided insights and direction for our design work with CoAware and helped lead to the aforementioned design suggestions. Its results also moved beyond prior literature (e.g., [44]) to allow us to more deeply understand when, where, and during what activities a system like CoAware would potentially be used in real-world situations. This can help guide future studies that want to test applications similar to CoAware to know when, where, and how such testing should be done. One could also imagine using Study 1's data on locations and context to think about ways to further refine applications such as CoAware. For example, users could be given options to customize applications so that they are able to choose what types of information they are comfortable revealing to their partner based on location, time, and activity. Such information could also be inferred by applications and then adjusted as needed by users.

We also recognize that there is a darker side to applications like CoAware and designers should be cautious in this regard. The challenge with apps that track or share mobile device usage between partners is that they can potentially alter relationship dynamics, given an increased access to information [13]. This could create issues around trust or control between partners. While our results did not reveal such concerns, they are most certainly possible. Further research is required to understand partners' information sharing behaviour with others and its impact on their relationship. We also acknowledge that apps with features like CoAware could be seen as being highly problematic for relationships that contain domestic abuse or family violence [14]. As apps like CoAware enable access to information on partners' devices, this could lead to the system being misused (e.g., coercing to share information constantly, surveilling one's partner) and create anxiety and tensions within a relationship. There are no easy solutions for such types of situations. CoAware could, for example, ask users for details about their relationship satisfaction before making features available to them. Yet partners in an abusive relationship could easily answer untruthfully. Apps could track couples' information sharing behaviours and provide warnings if acts that appear to resemble surveillance occur, or features could be turned off based on certain negative behaviours. However, this may, again, not be a complete solution and may be hard to detect. As such, designers need to be cautious to think about the possible negative consequences of apps with features similar to CoAware.

## 7 LIMITATIONS AND FUTURE WORK

Our crowdsourced survey has some inherent limitations. Since we used AMT, our participant's demographics were determined by the demographics (USA and India) of the ATM. With a larger sample and with participants from more cultures, it is possible to investigate how people's perceptions of co-located phone usage differ between cultures. It would also be interesting to conduct an in-person study with interviews to determine whether the results differ from those that have been obtained from crowdsourced study. In an in-person study, we would be able to see how people use the design for real-life situations which may not completely match the tasks in our study and essential to ground and guide the developments of CoAware. This would further help us cover ethical aspects of smartphone activity-awareness between couples which might be missed in our online survey. The challenges and potential problems that may arise with an increased awareness, especially in abusive and problematic relationships, needs further investigation. We believe such future research would provide us with important insights into the scope and impact (both positive and negative) of using CoAware and similar technological solutions in sensitive family situations.

We concentrated on partners' co-located smartphone activity awareness and investigated their opinions on CoAware. However, we envision extending our approach with CoAware to other relationship types, such as between parents and children, where the parent could use CoAware to monitor and control the child's smartphone activities. This would require an in-depth study of parent-children relationships and the consideration of many other aspects, such as the diversity of house rules, family traditions of raising children, child age, and the educational backgrounds of parents.

In the future, it would be interesting to find out in more detail how our findings were influenced by our participants' age, cultural background, and relationship length. This would require a larger and more diverse set of participant couples. Furthermore, with our initial encouraging results and reactions, we plan to further develop CoAware and to perform a longitudinal study with a full-fledged version to examine its effect on sustained behavior change among people. Additionally, CoAware inspires the design of context-aware smartphones where the solution can trigger notifications based on preset rules based on locations and any surrounding people to reduce the smartphone use of co-located persons.

## 8 CONCLUSION

Our paper examines the issues that arise when people use smartphones in front of their partners, how they try to solve these problems, and how they search for potential solutions. In a crowdsourced survey, we found that people often feel ignored and get frustrated due to their partner's smartphone use in front of them. We also found that these problems exist even though people consciously attempt to resolve them through mutual understanding, and sometimes with explicit house rules. Partners frequently ask about and share each other's smartphone activities verbally, yet are often not confident about knowing what the other is doing on the smartphone. Consequently, we designed CoAware, a smartphone app, to further explore the design space. CoAware allows partners to digitally share each other's smartphone activities using different levels of detail. In a study where couples used CoAware, we observed that they found the app to be a promising solution to improve awareness, help reduce smartphone overuse, and, perhaps, even monitor a child's smartphone activities. We learned that designing similar solutions requires careful consideration of various app-specific privacy concerns and people's tolerance towards being advised by their partners.

## 9 ACKNOWLEDGEMENTS

We thank all the participants for their time and feedback. This research was partially funded by a Natural Sciences and Engineering Research Council (NSERC) grant.

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
