# OpenReview forum: "CoAware: Designing Solutions for Being Aware of a Co-Located Partner’s Smartphone Usage Activities"
_graphicsinterface.org/Graphics_Interface/2021/Conference — GI 2021_

### Official Review · AnonReviewer1 · 2021-01-12
**CoAware: Designing Solutions for Being Aware of a Co-Located Partner’s Smartphone Usage Activities**

**Rating:** 7
**Confidence:** 3

**Review:**

This paper details thoughts, practices, habits, and perceptions of smartphone use in couples. This is done through two studies and a prototype; first habits, perceived wants, and other requirements are gathered from participants in in depth interviews via MTurk; a prototype based on this analysis is created; and a lab study is used to find out user acceptability and other initial feedback of their prototype.

pros:
- a motivated and fairly clear paper
- good use of in-depth qualitative analysis, which is more appropriate for this type of work
- good categories and stories led me along and helped me understand the data.
- descriptive statistics back up qualitative results to give more depth about the sample

cons:
- perhaps less impactful results - it's not clear how the results differ from current knowledge
- shallow discussion
- mismatched participant pools
- could have more method details for reproducibility

So overall I thought this paper was well done and informative. I think it's a strong work that provides the community with a body of insightful qualitative data that can be the basis for future work. I think the weakness is in the prototype and lab evaluation study which has questionable use and impact - this is admitted by the authors themselves. While the prototype on its own is fine, I think there was a missed opportunity for long-term evaluation. As this tool is meant primarily to be used in private spaces in private, romantic, long-term relationships, I think an "in-the-wild" study, even with just a few couples, would have been far more appropriate. As it stands, the lab study had good method and is acceptable, just that I feel the impact of those results is much less than the fascinating and in depth look into digital hygiene/habits in relationships. I think that the paper focused primarily on this first study was a good choice and enhances what I see as the the primary contribution.

Again on the positive note I think the qualitative analysis was well chosen, well organized, and the results well presented. I really feel like I got to understand the large amount of data generated in the MTurk study. In particular, I liked the means and averages and percentages given to back up or describe how common some sentiments were in the sample. My only small complaint, which may be due to length, was that there was not enough detail on how coders reconciled separate code bases, how much they differed, and how much they were similar. This would enhance the existing contribution. As it stands, I think it is okay to leave it out if space constrained (though a few extra sentences would be useful if it can be squeezed in), as even the same level of analysis from 1 coder would still make the results valuable to the community (though using 2, however well they matched, increases credibility).

I thought discussion was a bit shallow, again perhaps due to the amount of research and space available. I was hoping for more insight...I feel like just thinking about various typical user needs and the proposed solution could have generated many of the same insights. Of course, with the work in this paper it is data grounded and thus more reliable, so this is still valuable. I just normally look forward to a broader and deeper discussion, especially from such rich qualitative data.

A very small point is it was strange that the authors pulled 2 very culturally different countries as participants and did no cultural analysis. Especially as follow ups were done with exclusively NA participants. Views on family and socializing and rule-following are highly culture-sensitive, and while mentioned in the limitations, the exclusion felt very strange.

Thus, while there are drawbacks, I feel like the paper presents a reasonable and well done body of knowledge for the community, and is organized in a useful way that can act as a ground for further experiments and designs in this field. Thus, I would recommend acceptance.

Small points:
-new design motivation in related work felt out of place (see 2.3 e.g., we had granularity levels. This is new information and the real app design doesn't show up for many pages, so consider organization)
- Section 3 (first experiment) was initially unclear it was just a  requirements gathering experiment. This should be better clarified in the transition, as I was confused about what the experiment was about. Perhaps simply a better title for the section would improve e.g. "Exploration of Couple Smartphone Awareness"

---

### Official Review · AnonReviewer2 · 2021-01-13
**Review of CoAware: Designing Solutions for Being Aware of a C-located partner's smartphone usage activities**

**Rating:** 7
**Confidence:** 4

**Review:**

This paper is about the design of an app, called CoAware, that allows pairs to know about and see the usage of their partner's smartphone when they are physically co-located. The authors conducted an initial survey to understand people's willingness, interest and attitudes in having this knowledge, as well as their existing behaviour and practice in smartphone use during co-located activities such as during meals. The results of the survey were then used to inform the design of Co-Aware. An user evaluation of Co-Aware was conducted with 11 couples. The main findings of this evaluation were that people preferred knowing some detail about the type of use that was occurring, that notifications were intrusive and how much detail about smartphone use was depending on the type of application. There was also concern about having and maintaining some privacy about smartphone use.

This was a fun paper to read and the topic is intriguing. It will be of interest to the GI audience and will likely generate good conversations as most people have the experiences outlined in this paper. The authors used a well-designed methodology to create and evaluate CoAware. They discovered attitudes, practices and opinions from a wide audience via an on-line survey, and then motivated their app design based on the responses to the survey, followed by a user study.

However, there are some issues with this paper that reduced its rating:
1. Missing research objective/questions.
2. Do not report results in the introduction.
3. Once the common code set was created, how was the data then coded?
4. Why is it interesting that the younger participants thought it was important to have apps?
5. "We believe…." - Do not use believe here to explain a result like this? Re: privacy. The quote indicates embarrassment rather than sensitivity. Needs a reference from the literature.
6. While the descriptive statistics for the survey are interesting, there should be some statistical reporting given the quantity of data received. Statistics would provide a much stronger argument for some of the claims/conclusions in the paper.
7. "Improved awareness may help people use phones wisely." How is wisely measured? How is the quality of domestic life measured? These were not measured anywhere in this paper and should not be claimed here.
8. Discussion of survey is very limited and is not really a discussion of the survey results. I suggest combining results and discussion and answering the question “why did you get these results?”
9. Why was 30-sec used for closing an app? How was this time determined?
10. The theme table for study 2 is missing. What reliability method and measure was used?
11. Figure 5 is very confusing. The grey scale does not seem to match for the different areas.
12. Why would the authors use a non-parametric test for the within-subjects analysis and a parametric statistic for the between- subjects analysis. Need to report the normality statistics to justify the approach.
14. Discussion from study 2 needs support from the literature. The claims made do not necessarily link to the evidence. For example the claim made about personal digital space and Screen Share is not supported by the data. There may be other reasons why there was lower comfort ratings that I think would be closer. Some evidence from the literature would also be useful. A correlational statistic would also have been useful to look at those linkages rather than guessing.

Technical issues:
1. One question in Figure 1 is mis-worded, "how many hours do you send", I think should be "spend."
2. How was significance measured in Co-located Smartphone Use section, sentence 1
3. Grey scale in Figure 2 is difficult to see, use patterns instead
4. Writing is fairly sloppy and often colloquial (e.g., use of the verb “get”, words such as interestingly). A good editing pass is necessary.
5. Do not begin sentences with a number (e.g., 78%)
6. Figures with a, b, c, etc. on them are difficult to see. Need to revise format
7. Figures should always be located after their in-text reference, not before.
8. Using words such as “Of course, …..” should never be used in a formal academic research paper.
9. Need a statement about ethics approval for both studies.
10. There is some switching between past and present verb tenses in paragraphs making the paper difficult to read.

---

### Official Review · AnonReviewer3 · 2021-01-14
**Insight into issues and concerns around co-located smartphone usage between initimate partners**

**Rating:** 8
**Confidence:** 3

**Review:**

This paper presents detailed insight into issues and concerns that intimate cohabitating partners have about using smartphones while collocated. The in-depth qualitative analysis of a MTurk study provides a clear picture of the range and diversity of concerns, couples, and challenges. Following the paper presents a prototype developed from the results, and details a small in-lab study where people interact with the prototype.

Overall this is a very clearly written paper, with a solid standard trajectory from gaining insight using surveys, creating a prototype, and initial evaluation of the prototype. I quite enjoyed reading the stories and insights about how couples deal with the common smartphone use issue, and I think this data can be useful to the community (although a geographic comparison would have been nice). It is a model HCI paper and a strong candidate for GI this year.

Despite my enthusiasm for the survey results, on a whole unfortunately I found the paper to be lacking in maintaining that clear thread throughout. The app design is interesting, but I did not feel it was grounded sufficiently in the survey results. While the integration of personalization and privacy granularity was interesting, that could be derived from existing privacy literature. The inter-partner interaction particularly struck me as not grounded, for example the messaging: wouldn't existing messaging apps provide this feature? Wouldn't pre-designed messages come off as inauthentic? etc. As such the resulting app did not feel inspiring or all that novel.

Next, I was happy to see that the authors observed real people using their prototype, which can be useful for getting initial reactions and feedback. However, given how personal and contextual real use is, we have to admit simply how unrealistic an in-lab study is for this kin of work. As such, unfortunately I find little value in the statistical results as it is unclear to me if they can generalize. The initial positive reactions and potential concerns presented were useful, but I would have hoped for a much deeper treatment of this.

In the end, while this is a solid GI paper, it does have shortcomings that make me worry about the long term impact. My thoughts are that the paper would be a lot stronger were it streamlined. It is awfully verbose and seems to take tangents, where I think a more focused and honest treatment of the data collected (e.g., downplay / exclude stats from in-lab study, etc.) would result in a cleaner, and more strong, result.

---

### Meta-Review · Area_Chair1 · 2021-01-14

**Recommendation:** Accept
**Confidence:** 4

**Metareview:**

All reviewers seem to be in agreement  that the paper is in generally well written, and contains a valuable contribution in insights into smartphone usage, trust, and perception in couples. The main drawbacks are the lack of appropriateness and applicability of the lab study in part 3 of the paper. Reviewers also mentioned a weaker connection between the initial study, the prototype, and final user study.

Overall recommendations include better grounding the prototype design and research needs in the motivation of the 3rd user study, and focusing on improving some justifications and writing style (clarity, organizing, and prose are all mentioned by at least one reviewer). If extra space is needed, perhaps reducing the space dedicated to the in-lab experiment or its results may be prudent as all 3 reviewers raised concerns as to the generalizability and contribution of that section (though there is still value in it - we are not suggesting to remove it).

All reviewers, though, agreed that the paper provider a significant contribution in terms of insight into existing habits about smartphone use and provide good grounding qualitative data and proof-of-concept for future work. Thus, we recommend acceptance.

---

### Decision · Program_Chairs · 2021-01-16

Accept